# Low-Frequency Dielectric Relaxation in Structures Based on Macroporous Silicon with Meso-Macroporous Skin-Layer

**DOI:** 10.3390/ma14102471

**Published:** 2021-05-11

**Authors:** Rene Castro, Yulia Spivak, Sergey Shevchenko, Vyacheslav Moshnikov

**Affiliations:** 1Department of Physical Electronics, Faculty of Physics, Herzen State Pedagogical University of Russia (Herzen University), 191186 Saint Petersburg, Russia; recastro@mail.ru; 2Department of Micro- and Nanoelectronics, Faculty of Electronics, Saint Petersburg Electrotechnical University «LETI», 197376 Saint Petersburg, Russia; vamoshnikov@mail.ru; 3Department of Laser Measuring and Navigation Systems, Faculty of Information Measurement and Biotechnical Systems, Saint Petersburg Electrotechnical University «LETI», 197376 Saint Petersburg, Russia; syshevchenko@etu.ru

**Keywords:** porous silicon, impedance, interface, dielectric relaxation, low-frequency dielectric spectroscopy, ion-electron microscopy, temperature dependencies of dielectric permittivity

## Abstract

The spectra of dielectric relaxation of macroporous silicon with a mesoporous skin layer in the frequency range 1–10^6^ Hz during cooling (up to 293–173 K) and heating (293–333 K) are presented. Macroporous silicon (pore diameter ≈ 2.2–2.7 μm) with a meso-macroporous skin layer was obtained by the method of electrochemical anodic dissolution of monocrystalline silicon in a Unno-Imai cell. A mesoporous skin layer with a thickness of about 100–200 nm in the form of cone-shaped nanostructures with pore diameters near 13–25 nm and sizes of skeletal part about 35–40 nm by ion-electron microscopy was observed. The temperature dependence of the relaxation of the most probable relaxation time is characterized by two linear sections with different slope values; the change in the slope character is observed at T ≈ 250 K. The features of the distribution of relaxation times in meso-macroporous silicon at temperatures of 223, 273, and 293 K are revealed. The Havriliak-Negami approach was used for approximation of the relaxation curves *ε*″ = *f*(ν). The existence of a symmetric distribution of relaxers for all temperatures was found (Cole-Cole model). A discussion of results is provided, taking into account the structure of the studied object.

## 1. Introduction

One of the brightest trends in modern materials science is the creation of hierarchical porous materials, both organic and inorganic [1,2,3,4,5,6,7]. These materials are finding their application both independently and often as the 3D matrix media for complex multiphase compositions on their basis. The main areas of application of hierarchical porous materials and compositions based on them are power engineering (most often porous materials are used as electrodes of Li-ion batteries and fuel cells) [3,4,7,8,9,10], sensorics and biosensorics [11,12,13,14,15,16,17,18], catalysis [19,20,21], biotechnology, targeted drug delivery, and theranostics [20,21,22,23,24,25]. Another very promising area of application of such materials is bioelectronics, the element base for flexible wearable electronics [26,27,28,29,30,31]. For all these types of applications, the peculiarities of the interaction between the “matrix” and the “guest” and the processes taking place at their interface are the keys for obtaining and predicting the new properties and synergistic effects in such materials. In this case, at different hierarchical levels of pores, different effects can emerge and, so, different levels of pores can provide different functions [1,2,32,33,34,35,36,37]. The formation of porous fractal multicomponent systems with topology close to the infinitely coupled triply periodic minimal surfaces was sintered by sol-gel method [34]. In [36], a micro-meso-macroporous carbon material was obtained and activated by bacterial cellulose, with an open interconnected pore system and a specific surface area of 833 m^2^/g. The material showed improved characteristics in terms of the rate of mass transfer and adsorption capacity by the example of methylene blue in the volume of the pore system, as well as a significant improvement in catalytic properties during ethanol dehydration on the surface of a porous material. In [36], a three-dimensional hierarchical porous heterostructure In_2_O_3_/In_2_S_3_ with micro-meso-macropores was obtained, which showed a high photocatalytic activity. It is believed that such an effective photocatalytic ability is associated with the following aspects. First, the formation of In_2_O_3_/In_2_S_3_ heterostructures inhibits the recombination of photogenerated carriers. Second, a hierarchically porous structure acting as a micro-nano-reactor can have a large surface area and many reactive sites. Third, open 3D scaffolds can increase light collection and facilitate mass transfer of reagents. In [37], meso-macroporous silicon is considered, in which vertical macropores (130–140 nm in diameter) can act as channels for the supply/removal of reagents and reaction products. In this case, the system of side branches—mesopores with sizes of the order of 30–40 nm and increasing the area of the inner surface of por-Si—could play the role of nanoreactors and catalyst carriers. Porous materials of this kind are promising for various types of sensors, catalytic membranes, and fuel cell electrodes.

One of the sensitive and informative methods for studying such complex low-dimensional systems is impedance spectroscopy in a wide frequency range [38,39,40,41,42,43]. Especially important for the study of electronic processes, possible migration of ions, etc., is impedance spectroscopy at low frequencies in the presence of a constant component of the electric field.

Macroporous silicon with a columnar type of texture is often considered promising for application in various types of devices: as antireflection coatings in the solar cells, as a matrix material functionalized by nanomaterials for gas sensors of various types, as well as a media for the low-threshold emission electrodes, etc. An important advantage of using por-Si is the possibility of integration with elements of silicon microelectronics into a union chip. For such applications, the electrophysical properties of por-SI and its dependencies on temperature and frequency are important. It was found from previous studies that macroporous p-type silicon with a columnar texture and a skin layer shows a complex frequency dependence of the capacitance, as well as the presence of a maximum dielectric loss tangent, which are most likely associated with the dipole relaxation mechanism of polarization [44]. This work is a further development of the experimental study of the dielectric properties of macroporous silicon with a mesoporous skin layer by the method of impedance spectroscopy in the region of ultralow frequencies.

## 2. Materials and Methods

To obtain porous silicon, p-type monocrystalline silicon wafers with a resistivity of 12 Ohm*cm and a crystallographic orientation (100) were processed by the method of electrochemical anodic etching. The laboratory setup based on Unno-Imai type of electrochemical cell was used [45,46]. Electrical contact with the cathode side of the silicon wafer is made by contacting the Si with a low resistance electrolyte. This provides high uniformity of the characteristics of the porous layer over the wafer area. The procedure was as follows: a silicon wafer was fixed in a holder-membrane of a polypropylene bath, dividing it into two independent volumes. Platinum electrodes were installed on both sides of the plate at the same distance. The bath was filled with electrolyte. An aqueous solution of hydrofluoric acid with the addition of isopropanol was used as the electrolyte. The electrodes were connected to a power source, and the silicon was processed in the current stabilization mode. Etching was carried out at an anodic current density of 30 mA/cm^2^ for 10 min. A por-Si layer was formed on the anodic side of the Si plate. Technological conditions were chosen for the formation of the columnar macroporous silicon with mesoporous surface layer (the so-called “skin” layer), which usually is heterophase.

Studies of the morphology and structure of porous silicon were carried out using ion-electron microscopy (Helios Nanolab D449 (FEI Company). The studies were carried out at an accelerating voltage of 5–10 kV. The surface was observed in a magnification range of ×8000–×100,000 times at an angle of 52° to obtain information simultaneously on the morphology of the cleavage and the surface. Quantitative measurements of geometric parameters were carried out, taking into account the angle of observation of the surface.

Measurements of the frequency dependences of the dielectric coefficients of por-Si layers were performed using a Concept-41 spectrometer from Novo-control Technologies in the frequency range 1 < f < 10^6^ Hz in two temperature ranges: during cooling (293–173 K) and heating (293–333 K). The setup consisted of a frequency impedance analyzer, a measuring cell, and an automatic data acquisition system with a computer interface. To carry out measurements, contacts were formed on the surface of porous silicon using a silver paste, followed by drying.

## 3. Results and Discussion

### 3.1. Microstructure Characterization

According to the data of electron-ion microscopy, it was found that under the selected technological conditions, porous silicon with a complex type of texture is formed, which is a macroporous layer covered with a porous surface layer (“skin”—layer) (Figure 1). An overview of the cleavage and surface of porous silicon is shown in Figure 1a: a macroporous layer with a p-Si (100) type of pores close to regular shape covered with a mechanically stressed skin layer. Herewith, the texture reveals two levels of macropores: in addition to the aforementioned macroporous columnar pores with diameter ≈ 2.2–2.7 μm (“macropore 1”), macropores are observed within a mesoporous skin layer (“macropore 2”). Figure 1b shows that a mesoporous layer is found on the walls of macropores 1. Additionally, one can find from Figure 1a,b, macropores of type 1 are interconnected by channels to each other. A mesoporous skin layer with a thickness of about 100–200 nm in the form of cone-shaped nanostructures with pore diameters near 13–25 nm and sizes of skeletal part about 35–40 nm was observed (Figure 1c,d). In addition, the meso-macroporous layer has a higher resistance compared to the skeletal part of macroporous silicon.

This is due to the effect of intense accumulation of negative charge on the path of primary electrons, that is, deterioration of the conditions for the release of secondary electrons (in fact, a barrier field is formed) creates a change in the image contrast—the so called “charge” of local surface areas. The presence of such effects in Figure 1c,d made it possible to conclude that this layer is highly resistive.

The geometric characteristics of por-Si are presented in Table 1.

The observed type of macropores 1 with a skin layer on the surface is typical for porous silicon formed in high-resistance single-crystal p-Si (100) [37,47,48,49]. The formation of a skin layer in such structures is associated with the removal and redeposition of reaction products during electrochemical anodization during the formation of porous silicon, taking into account the redistribution of the electric field strength over the surface of the growing porous layer. In our previous studies, the phase composition of the “skin” layer is complex, consisting mainly of a mixture of SiO_2_, SiO_x_, and also containing αSi: H, nanocrystals of secondary recrystallized silicon, etc. [37,45,50,51]. A distinctive feature of porous silicon obtained in this work is that the “skin” layer is meso-macroporous, while its surface is covered with cone-shaped nanostructures. The formation of this type of texture was most likely influenced by two main factors—the intense removal of reaction products during the formation of the main macropores (this causes the appearance of openings in skin layer-macropores of type 2) and the redistribution of electric field strength on the nano protrusions, which contributes to the intensive redeposition of the reaction products.

Such a mesoporous layer can fulfill an important practical function. Such a porous silicon texture can be used as a template for the growth of nanostructured materials on a surface with the further formation of an emission material [52]. A high-resistance skin layer is effective for por-Si intercalation processes, for example, when metals are introduced into pores electrochemically, since it promotes the concentration of the electric field at the bottom of macropores 1 [47].

In addition, macro-mesoporous structures of this kind are interesting as growth platforms, optically inhomogeneous media, biosensors, as implant materials, etc. [53,54,55].

### 3.2. Impedance Spectroscopy Characterization

The measurements of the frequency dependences of the dielectric coefficients of the por-Si layers were performed using a Concept-81 spectrometer in the frequency range 1 < *f* < 10^6^ Hz with a voltage of 1 V. The frequency dependences of the dielectric permittivity *ε*′ of por-Si layers in the temperature range 283–333 K are shown in Figure 2. Figure 2 demonstrates a complex character of the *ε*′ dispersion via frequency, which correlates with our previous studies [44]. At the low frequencies, a region of dielectric permittivity *ε*′ values is weakly dependent on frequency (frequency range (1 Hz < *f* < 10 Hz), followed by a significant decrease in *ε*′ values in the range of 10 Hz < *f* < 10^3^ Hz. The dielectric permittivity decreases by 50%. This dependence indicates the manifestation of the dipole-relaxation type of polarization in the low-frequency region. Starting from values of the order of 3 × 10^3^ Hz and higher, *ε*′ becomes independent of the frequency of the alternating field, which is associated with the ordering of the dipole molecules in the direction of the electric field. A similar dependence of the capacitance on frequency was obtained in [56] for a structure with a p-por-Si layer obtained under similar technological conditions, where the presence of two polarization mechanisms is also revealed.

The temperature dependence of the dielectric permittivity *ε*′ demonstrates the following features. As can be seen from Figure 2, an increase in the dielectric permittivity *ε*′ of por-Si is observed with increasing temperature. This also confirms the dipole-relaxation type of polarization, in which an increase in temperature promotes the rotation of the dipoles in accordance with the applied field. The temperature dependence of *ε*′ for a frequency of 10^3^ Hz shows the presence of two parts with different rates of increase in the dielectric constant (Figure 3): at a temperature of about 305 K, a change in the nature of the dependence is observed, and with a further rise in temperature, the rate of increase of *ε*′ significantly grows. These results are in agreement with the conclusions about the polarization mechanism stated above.

Figure 4 presents the frequency dependence of the dielectric loss factor *ε*″ of por-Si layers in the temperature range T = 193–293 K. 

As can be seen from Figure 4, the frequency dependence of *ε*″ has an explicit maximum, the position of which depends on temperature: at room temperature, the maximum *ε*″ is found at a frequency of ≈ 200 Hz, and upon cooling it shifts to the region of lower frequencies (up to 1.5–2 Hz at 193 K). This character of dependence is typical for the dipole polarization mechanism, when the dielectric loss factor increases until the relaxation time of the dipole molecules reaches the value τ = 1/2 *f*, then the *ε*″(*f*) decays.

Figure 5 shows the temperature dependence of the relaxation times of the most probable relaxation time *τ*_max_ for por-Si layers. The temperature dependence of the *τ*_max_ reveals the presence of two temperature regions with activation energies *E_A_*_1_ = 0.25 eV and *E_A_*_2_ = 0.08 eV. The transition from one section to another is observed at the same critical temperature at which there is a maximum of the dielectric loss factor *ε*″.

The relaxation curves *ε*″ = *f*(ν) were approximated within the Havriliak-Negami approach [57,58,59]:(1)$$\varepsilon =\varepsilon_{\infty }+\frac{\varepsilon_{s}-\varepsilon_{\infty }}{{\left(1+{\left(i\omega \tau \right)}^{1-\alpha }\right)}^{\beta }}$$
where *ε* is the dielectric permittivity, *ω* is the frequency, $\varepsilon_{\infty }$ is the dielectric permittivity at *ω* → ∞, *τ* is the relaxation time, and *α* and *β* are some parameters of frequency dispersion. The Havriliak-Negami approach is often considered as a generalization of previous models for the frequency dispersion of the dielectric permittivity [59]:for *α* = 0 and *β* = 1, Formula (1) transforms into the Debye model;for 1 > *α* > 0 and *β* = 1—the Cole-Cole model;for *α* = 0 and 1 > *β* > 1—Formula (1) turns into the Davidson-Cole model.

The approximation of the relaxation curves *ε*″ = *f*(ν) using the Havriliak-Negami approach made it possible to reveal the existence of a distribution of relaxers. Wherein, for all temperatures, a symmetric distribution of relaxers over relaxation times is obtained, which is in agreement with the Cole-Cole model.

Note that for p-por-Si (100), the study of the temperature dependence of the conductivity was mainly published, and a change in the nature of the conductivity in a close temperature range was also found [56,60,61,62]. Temperatures below the inflection point are characterized by the hopping type of conductivity in such structures. In this case it is possible to implement the mechanism of hopping conductivity with a variable hopping length over a three-dimensional system of localized states [60], which is in good agreement with the texture characteristics for the porous material studied in this work. In [61], the electrical properties of porous silicon nanoparticles distributed in a dielectric matrix (epoxy resin) are considered, and the transition is observed at close temperatures. Taking into account the microstructure of the studied porous silicon, it should be noted that in such a material many different systems can act as relaxers: nanosized oxide regions, various defects in the structure and at the interface, functional groups, etc. Their identification requires further research. This is in agreement with our previous studies of the structure and phase composition of similar layers of porous silicon [37,45,50,51], showing that the skin layer consists of more than 60% SiO_2_ and SiO_x_ with embedded Si nanocrystals and also contains αSi:H.

Figure 6 shows the shape of the function of the relaxation time distribution. Note that before and after the critical temperature T = 250 K, the form of the function indicates a change in the energy spectrum of the por-Si. This change can be associated with the ejection of electrons from trapped states at a certain temperature.

One can assume that with an increase in temperature, more types of relaxers can participate in the process. Thus, the system consists of several different types of relaxers. As a result of the presence of a spectrum of localized states due to the defectiveness of the structure, one can assume the existence of electrically active defects.

## 4. Conclusions

The results of a study of the temperature dependence of dielectric relaxation in the low-frequency region in macroporous silicon with a meso-macroporous skin layer are presented. The por-Si texture parameters were determined by ion-electron microscopy: columnar macroporous silicon with pore diameter of ≈2.2–2.7 μm covered by a meso-macroporous skin layer. A meso-macroporous skin layer with a thickness of about 100–200 nm is in the form of cone-shaped nanostructures with mesopore diameters near 13–25 nm and macropore about 190–310 nm. Sizes of skeletal part of meso-macroporous skin-layer are about 35–40 nm. 

A complex dispersion of the dielectric permittivity *ε*′ was observed in the investigated frequency range. The presence of a maximum of the loss factor *ε*″ was found, which shifts in the region of higher frequencies. This indicates the existence of a relaxation process in the investigated temperature range. The temperature dependence of the most probable relaxation time *τ*_max_ showed the presence of two temperature regions with activation energies *E_A_*_1_ = 0.25 eV and *E_A_*_2_ = 0.08 eV. The approximation of the relaxation curves *ε*″ = *f*(ν) using the Havriliak-Negami approach made it possible to ascertain the existence of a symmetric distribution of relaxers for all temperatures (Cole-Cole model). The shape of the distribution function of relaxation times before and after the critical temperature T = 250 K indicates a change in the energy spectrum of the por-Si. This change can be associated with the ejection of electrons from trapped states at a certain temperature. The presence of a spectrum of localized states due to the defectiveness of the structure may suggest the existence of electrically active defects. 

The temperature and frequency dependencies on dielectric relaxation in por-Si obtained in this work will be useful in the development of integrated functional devices based on porous silicon, such as elements of solar cells, PETE electrodes, low-field emission electrodes, biosensors and integrated gas sensors, optical filters, and so on. Further study of this kind of materials involves considering their properties taking into account the fractal structure of porous silicon and the possible existence of several self-similar relaxing subsystems in such a complex multiphase object.

## Figures and Tables

**Figure 1 materials-14-02471-f001:**
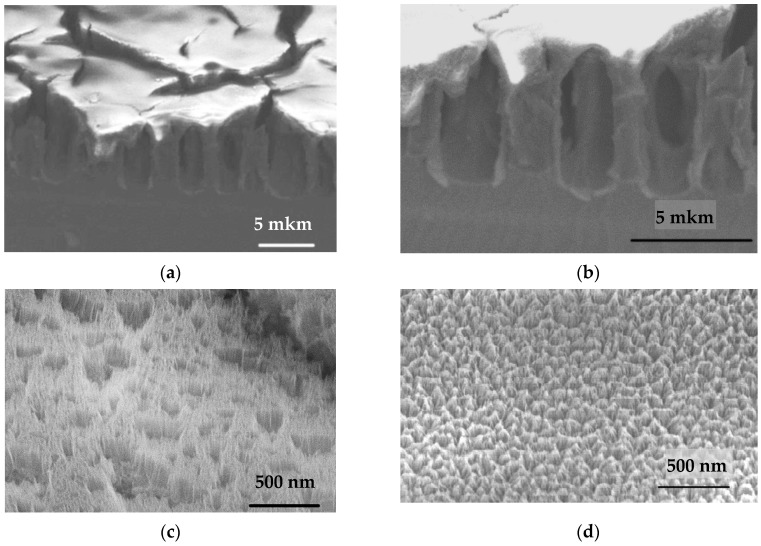
The microstructure of porous silicon according to ion-electron microscopy data (**a**,**b**) is a general view of the por-Si and surface at various magnifications (×8000 and ×15,000 times, respectively); (**c**) the view of the skin layer (presumably) over the macropore type 1; (**d**) view of the skin layer (presumably) above the skeletal part of the macroporous layer, cone-shaped porous nanostructures are visible; (**e**) a schematic representation of the sample microstructure: 1—Si; 2—macropore 1; 3—macropore 2; 4—cone-shaped nanostructures; 5—mesoporous coating.

**Figure 2 materials-14-02471-f002:**
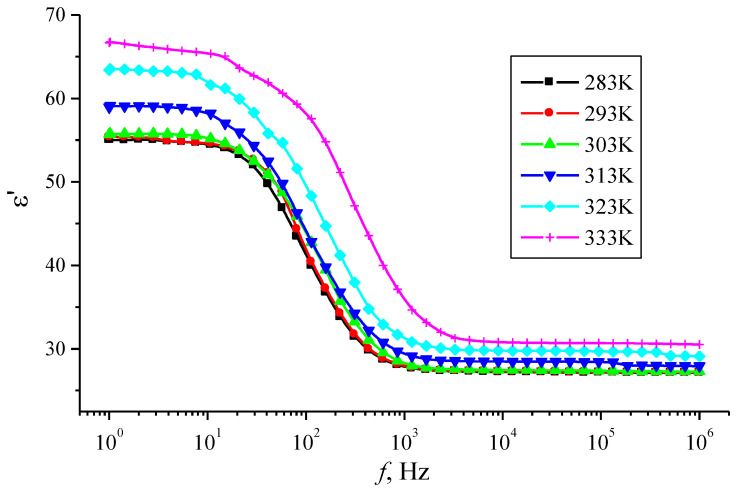
Frequency dependence of the dielectric constant *ε*′ of por-Si layers at different temperatures.

**Figure 3 materials-14-02471-f003:**
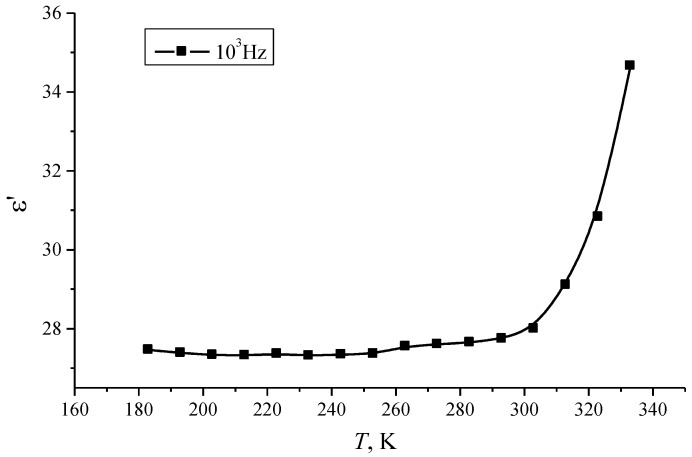
Temperature dependence of the dielectric constant *ε*′ of por-Si layers at a frequency of *f* = 10^3^ Hz.

**Figure 4 materials-14-02471-f004:**
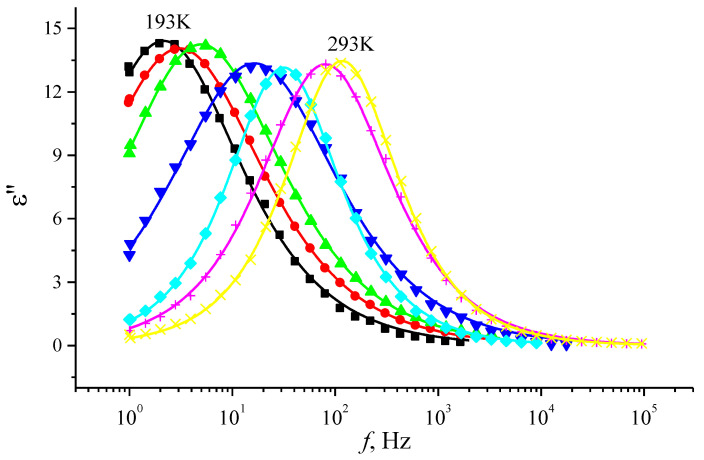
Frequency dependence of the dielectric loss factor *ε*″ of por-Si layers at different temperatures: T = 193 K (black line), 203 K (red line), 213 K (green line), 223 K (blue line), 233 K (light blue line), 272 K (purple line), 293 K (yellow line).

**Figure 5 materials-14-02471-f005:**
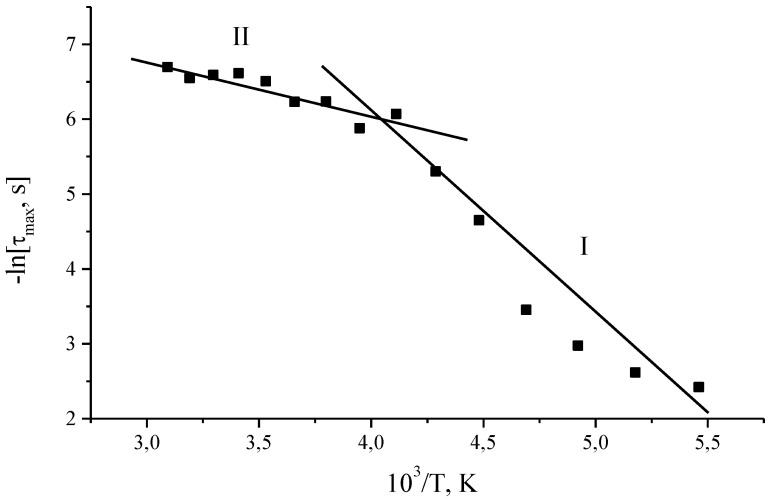
Temperature dependence of the relaxation times of the most probable relaxation time *τ*_max_ for por-Si layers. The graph reveals two sections with different slope (different activation energies): section I with estimated activation energy *E_A_*_1_ = 0.25 eV and section II with estimated activation energy *E_A_*_2_ = 0.08 eV.

**Figure 6 materials-14-02471-f006:**
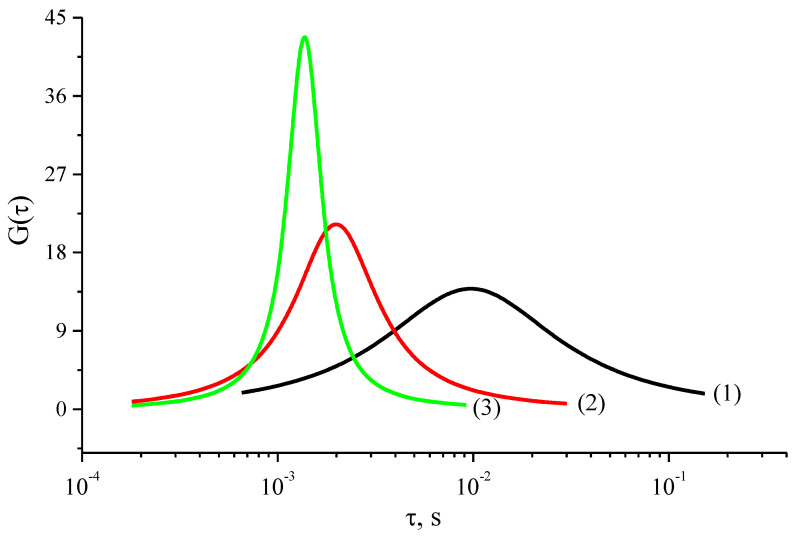
Form of the relaxation time distribution function for por-Si layers at different temperatures T = 223 K (curve (1)), 273 K (curve (2)), 293 K (curve (3)).

**Table 1 materials-14-02471-t001:** Parameters of meso-macroporous silicon texture.

Geometric Elements of Por-Si Texture	Thickness, µm	Pore Diameter, nm	Size of Skeletal Part, nm
Macropores 1	7.8–8.2	2200–2700	0600–1200
Macropores 2	0.1–0.2	190–310	-
Mesopores	0.1–0.2	13–25	35–40

## Data Availability

The data presented in this study are available on request from the corresponding author.

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
