# Peer review of "Low-Frequency Dielectric Relaxation in Structures Based on Macroporous Silicon with Meso-Macroporous Skin-Layer"

_materials, 2021, doi:10.3390/ma14102471_

Round 1

Reviewer 1 Report

In this paper, the author reported low-frequency dielectric relaxation in structures based on macroporous silicon with meso-macroporous skin-layer. The spectra of dielectric relaxation of macroporous silicon with a mesoporous skin layer in the frequency range 1-106 Hz in during cooling (up to 293-173 K) and heating (293-333 K) are presented. The work is innovative and the research is systematic. It is suggested to solve the following problems before publishing.

  1. Summary part of the author needs to be concise, the article is too cumbersome.
  2. I suggest that figure 1 be deleted. The process of electrochemical etching of silicon is relatively mature, so it is unnecessary for the author to draw a schematic diagram.
  3. The author should make a table to compare the advantages of the work.

Author Response

Dear reviewer,

Thank you very much for your comments and suggestions, and for a detailed analysis of our manuscript. We have corrected the manuscript taking into account all the comments made.

All corrections are highlighted in green. Because one figure was excluded, the other figures and the table had to be moved so that they fit well in the text. Because of this, the line numbers in the comments of the reviewers and in the new version of the manuscript may not coincide (therefore, in the responses to the reviewers, we indicated new lines to the comments).

The answer for Reviewer 1

  1. To the comment: “Summary part of the author needs to be concise, the article is too cumbersome”.

Answer:

The Summary was corrected and shortened in accordance with the recommendation of the reviewer 1 first comment as far as possible without losing the findings.

  1. To the comment: “I suggest that figure 1 be deleted. The process of electrochemical etching of silicon is relatively mature, so it is unnecessary for the author to draw a schematic diagram.”

Answer:

At the request of reviewer 1 (second comment), Figure 1 was excluded. The process of porous silicon formation by the method of electrochemical anodic etching, including the Unno-Imai method are indeed well known. The authors considered it necessary to include the figure, since the technology of obtaining porous silicon is a multifactorial process, this leads to a wide variety of textures of porous silicon, and it is good to indicate the specific conditions for obtaining, so to make it clear what kind of porous silicon is discussed in this work. But we agreed to exclude Fig. 1.

  1. To the comment: “The author should make a table to compare the advantages of the work”.

Answer:

Noting the advantages of the work, we would like to highlight the following:

  • The results of studies of dielectric relaxation in a wide frequency range and a wide temperature range (both during cooling and heating) are new. According to the literature known to the authors, this material has not been studied in such ranges.
  • Studies in a wide frequency range, especially in the low-frequency range, made it possible to identify the dipole relaxation mechanism in porous silicon with this type of texture.
  • Temperature studies of the relaxation time distribution function revealed a change in the energy spectrum of por-Si before and after the critical temperature. In this work, an attempt is made to interpret it considering the microstructure of the material. It is assumed that the change in the energy spectrum can be associated with the release of electrons from trapped states at a certain temperature. The presence of a spectrum of localized states due to the defectiveness of the structure may indicate the existence of electrically active defects.
  • The experimental results obtained in this work are important in the development of device structures for various purposes, which include macroporous silicon. The type of macroporous p-silicon with a well-type texture investigated in this work is also relevant from a practical point of view, both as an independent element and as part of hybrid nanocomposites. It is used to create gas sensors of various operating principles, including gas-sensitive hybrid rod nanostructures based on silicon with branched metal oxide coatings, biosensors and systems for targeted drug delivery and theranostics, solar cells and so on [for example: Solid-State Electronics 2018, https://doi.org/10.1016/j.sse.2018.12.011; Status Solidi A 2017, 1700565 https://doi.org/10.1002/pssa.201700565; Microporous and Mesoporous Materials 2019, https://doi.org/10.1016/j.micromeso.2019.109619].

Yulia Spivak

Reviewer 2 Report

Overall, the manuscript looks solid and well-organized. The reviewer would like to comments on a few minor things. First, Fig 2 needs to have zoomed-in images. For example, Fig2. (d) has written dimensions but these are really hard to read. Therefore, zoomed-in images with proper legends would be helpful. Second, there are a few grammar errors and a revision needs to be conducted. For example, line 149, ".... by electrochemically..." should be ...electrochemically...".  Finally, it would be informative if the authors can elaborate potential applications of the findings in the conclusion section.    

Author Response

Dear reviewer,

Thank you very much for your comments and suggestions, and for a detailed analysis of our manuscript. We have corrected the manuscript taking into account all the comments made.

All corrections are highlighted in green. Because one figure was excluded, the other figures and the table had to be moved so that they fit well in the text. Because of this, the line numbers in the comments of the reviewers and in the new version of the manuscript may not coincide (therefore, in the responses to the reviewers, we indicated new lines to the comments).

The answer for Reviewer 2

  1. To the comment: “First, Fig 2 needs to have zoomed-in images. For example, Fig2. (d) has written dimensions but these are really hard to read. Therefore, zoomed-in images with proper legends would be helpful.”

Answer:

The Figure 2 - the morphology and structure of porous silicon by ion-electron microscopy (after excluding the Fig. 1, in new numeration this figure became Fig. 1): all parts of Fig.  were enlarged and the legends were corrected and supplemented.

  1. To the comment: “Second, there are a few grammar errors, and a revision needs to be conducted. For example, line 149, ".... by electrochemically..." should be ...electrochemically...".

Answer:

Corrections have been made (new line number 147).

  1. To the comment: “Finally, it would be informative if the authors can elaborate potential applications of the findings in the conclusion section.”

Answer:

The conclusions were supplemented with information on possible applications of the results of the work. The temperature and frequency dependencies on dielectric relaxation in por-Si obtained in this work will be useful in the development of integrated functional devices based on porous silicon, such as elements of solar cells, PETE electrodes, biosensors and integrated gas sensors, optical filters and so on.

Yulia Spivak

Reviewer 3 Report

Very well organized and structured research work on low-frequency dielectric relaxation in macroporous silicon-based structures with mesomacroporous skin-layer.
Manuscript prepared with IMRaD structure, it is easy to read, and the main research flow is clear. 
The title of the manuscript is informative and relevant. Abstract match the rest of the article, but some thoughts about the discussion of the results are missing. Please improve the abstract.
The introduction section is well organized. It starts from the general and goes towards the specificities of the field. But the main aim of the investigation and motivation are not presented clearly. Please improve this section. 
Study methods are valid and reliable. The used units are appropriate. Tables and figures are relevant and clearly presented. Only one remark about the figures' quality: The green texts on Fig.1 b) d) are not readable. Please use other colors.

Some comments for the discussion sections:
-Row 172: The authors write that dipole relaxation is decreasing with temperature. Arguing with the authors, susceptibility due to the orientation relaxation proportional to the inverse temperature; hence the permittivity has to also decrease with increasing temperature. I think this relaxation is related to increasing conductivity. The results of Fig. 7 also support it, and the authors also state it in rows 209-230. Please think it through. 

-Fig 6. My feeling is that the approximation of the measurement data with two linear sections is a  bit sketchy. The data suggests three straight lines for approximation. Line II looks appropriate for the 3.0...4.0 1000/K. The line 'I' has to be separated into two sections: one section for 4.0...4.7 1000/K, the second one for 4.7....5.5 1000/K range. Please think it through. 

Author Response

Dear reviewer,

Thank you very much for your comments and suggestions, and for a detailed analysis of our manuscript. We have corrected the manuscript taking into account all the comments made.

All corrections are highlighted in green. Because one figure was excluded, the other figures and the table had to be moved so that they fit well in the text. Because of this, the line numbers in the comments of the reviewers and in the new version of the manuscript may not coincide (therefore, in the responses to the reviewers, we indicated new lines to the comments).

The answer for Reviewer 3

  1. To the comment: “Abstract match the rest of the article, but some thoughts about the discussion of the results are missing. Please improve the abstract.”

Answer:

Information on the interpretation of the results has been added to the abstract.

  1. To the comment: “But the main aim of the investigation and motivation are not presented clearly. Please improve this section”.

Answer:

Corrections have been made in Introduction and the aim sections.

  1. To the comment: “Only one remark about the figures' quality: The green texts on Fig.1 b) d) are not readable. Please use other colors.”

Answer:

Corrections have been made. We removed the dimensions on the figures, which were difficult to read. Information about the characteristic geometrical dimensions is given in the text of the manuscript and in the table.

  1. To the comment: “Row 172: The authors write that dipole relaxation is decreasing with temperature. Arguing with the authors, susceptibility due to the orientation relaxation proportional to the inverse temperature; hence the permittivity has to also decrease with increasing temperature. I think this relaxation is related to increasing conductivity. The results of Fig. 7 also support it, and the authors also state it in rows 209-230. Please think it through.”

Answer:

After a detailed discussion of the results, taking into account the remarks of reviewer 3, the authors came to the following.

As we wrote in manuscript, a complex dispersion of the dielectric permittivity ε' is observed in the investigated frequency range. At the low frequencies, a region of dielectric permittivitye' values is weakly dependent on frequency (frequency range (1 Hz < f <10 Hz), followed by a significant decrease in ε' values in the range of 10 Hz < f <103 Hz. This dependence indicates the manifestation of the dipole-relaxation type of polarization in the low-frequency region. Starting from values of the order of 3⸱103 Hz and higher, ε' becomes independent of the frequency of the alternating field, which is associated with the ordering of the dipole molecules in the direction of the electric field.

With increase of the temperature the ε' increasing. This also confirms the dipole-relaxation type of polarization, in which an increase in temperature promotes the rotation of the dipoles in accordance with the applied field.

  1. To the comment: “Fig 6. My feeling is that the approximation of the measurement data with two linear sections is a bit sketchy. The data suggests three straight lines for approximation. Line II looks appropriate for the 3.0...4.0 1000/K. The line 'I' has to be separated into two sections: one section for 4.0...4.7 1000/K, the second one for 4.7....5.5 1000/K range. Please think it through.”

Answer:

We carefully considered the comments and discussed the results again. In our opinion, it is not necessary to highlight the third section on the temperature dependence of relaxation time τmax. There are two intervals: fast change (lower T) and slow change (higher T). There is confirmation from the literature about the existence of one critical transition point in this temperature range. To select the third interval, it is necessary to carry out additional measurements at low temperatures with a smaller step of changing this parameter and using other experimental techniques.

Yulia Spivak

Round 2

Reviewer 1 Report

The article has been modified systematically and can be accepted.

Author Response

Response to Reviewer 1 Comments

We have changed the form of responses to Reviewer comments according to the recommended example. Also, in response to point 3, we gave full references to literature sources, which we cite as examples.

Point 1: “Summary part of the author needs to be concise, the article is too cumbersome”.

Response 1: The Summary was corrected and shortened. We've reworked and removed the small details from the first paragraph that relate to the technology and parameters of the porous texture not very significant for summary, keeping the main points (lines 250-254). Also, lines 273-275 were removed, as it was discussed already in the Results and Discussion part. We had to add lines 276-279 to the Summary according to the comment of the reviewer, who asked to add possible applications of the obtained results in practice to Summary part.

Point 2: “I suggest that figure 1 be deleted. The process of electrochemical etching of silicon is relatively mature, so it is unnecessary for the author to draw a schematic diagram.”

Response 2: At the request of reviewer 1 (second comment), Figure 1 was excluded. The process of porous silicon formation by the method of electrochemical anodic etching, including the Unno-Imai method are indeed well known. The authors considered it necessary to include the figure, since the technology of obtaining porous silicon is a multifactorial process, this leads to a wide variety of textures of porous silicon, and it is good to indicate the specific conditions for obtaining, so to make it clear what kind of porous silicon is discussed in this work. But we agreed to exclude Fig. 1. The text was corrected accordingly (the reference to Fig. 1 with the setup diagram was removed and references were made to articles where this type of electrochemical cells were used) – lines 85-87. Figure 1 and caption was also removed – lines 103-107.

Point 3: “The author should make a table to compare the advantages of the work”.

Response 3: Noting the advantages of the work, we would like to highlight the following:

  • The results of studies of dielectric relaxation in a wide frequency range and a wide temperature range (both during cooling and heating) are new. According to the literature known to the authors, this material has not been studied in such ranges.
  • Studies in a wide frequency range, especially in the low-frequency range, made it possible to identify the dipole relaxation mechanism in porous silicon with this type of texture.
  • Temperature studies of the relaxation time distribution function revealed a change in the energy spectrum of por-Si before and after the critical temperature. In this work, an attempt is made to interpret it considering the microstructure of the material. It is assumed that the change in the energy spectrum can be associated with the release of electrons from trapped states at a certain temperature. The presence of a spectrum of localized states due to the defectiveness of the structure may indicate the existence of electrically active defects.
  • The experimental results obtained in this work are important in the development of device structures for various purposes, which include macroporous silicon. The type of macroporous p-silicon with a well-type of texture investigated in this work is also relevant from a practical point of view, both as an independent element and as part of hybrid nanocomposites. It is used to create gas sensors of various operating principles, including gas-sensitive hybrid rod nanostructures based on silicon with branched metal oxide coatings, biosensors, and systems for targeted drug delivery and theranostics, solar cells and so on. References to some examples:
  • Alwan M. Alwan, Duaa A. Hashim, Muslim F. Jawad, Efficient bimetallic nanoparticles embedded-porous silicon CO gas sensor, Solid-State Electronics, 2019. Volume 153. Pages 37-45. https://doi.org/10.1016/j.sse.2018.12.011 .
  • Georgobiani V.A., Gonchar K.A., Zvereva E.A., Osminkina L.A., Porous Silicon Nanowire Arrays for Reversible Optical Gas Sensing, Phys. Status Solidi A 2017, 1700565 https://doi.org/10.1002/pssa.201700565;
  • Mihir Kumar Sahoo, Paresh Kale, Restructured porous silicon for solar photovoltaic: A review, Microporous and Mesoporous Materials, 2019. Volume 289. 109619. https://doi.org/10.1016/j.micromeso.2019.109619.
  • Le Borgne, L. Pichon, A. C. Salaun et al. Bacteria electrical detection using 3D silicon nanowires based resistor // Sensors & Actuators: B. Chemical 273 (2018) 1794–1799. https://hal-univ-rennes1.archives-ouvertes.fr/hal-01860675/file/Manuscriptvf.pdf

Yulia Spivak

Reviewer 3 Report

Thank you for considering my comments. The paper has been improved. Tell the truth, I still argue with the statement of the authors' answer for my comment 4. 

The authors also write in the paper and the answer: "With increase of the temperature the ε' increasing. This also confirms the dipole-relaxation type of polarization, in which an increase in temperature promotes the rotation of the dipoles in accordance with the applied field."

This statement is not true. In dipole polarization, the electric field tries to orient the dipoles parallel to the field. Thermal energy counteracts this tendency. With increasing temperature, the latter becomes stronger. Hence the polarizability becomes lower. In general, the inverse temperature dependence is the most trivial evidence of the presence of dipole polarization. I believe the authors experienced a more complex phenomenon.

Author Response

Response to Reviewer 3 Comments

Point 1: “Tell the truth, I still argue with the statement of the authors' answer for my comment 4.

The authors also write in the paper and the answer: "With increase of the temperature the ε' increasing. This also confirms the dipole-relaxation type of polarization, in which an increase in temperature promotes the rotation of the dipoles in accordance with the applied field."

This statement is not true. In dipole polarization, the electric field tries to orient the dipoles parallel to the field. Thermal energy counteracts this tendency. With increasing temperature, the latter becomes stronger. Hence the polarizability becomes lower. In general, the inverse temperature dependence is the most trivial evidence of the presence of dipole polarization. I believe the authors experienced a more complex phenomenon.”.

Response 1: We totally agree with Reviewer 3 that the phenomenon of dielectric polarization in such a complex heterophase material as porous silicon is of a much more complex nature than when considering the classical mechanisms of polarization. In contrast to the dipole-relaxation component of polarization in a homogeneous bulk material, in an inhomogeneous heterostructure, such as porous silicon, dipole-relaxation type of polarization is more complex and specific. Dipoles at phase boundaries cannot freely pass from one state to another (cannot rotate freely) with alternating electrical field due to the influence of heterointerfaces. It is expected that the nature of losses in such systems will differ from the classical one.

Possible manifestations of interlayer polarization, the influence of active defects, for example, due to nonstoichiometric silicon oxide in the composition of the skin layer and / or hydronium ions. Complex behavior can also lead to the manifestation of inductance associated with the fractal-percolation structure of the layer [Pronin, I. A., Yakushova, N. D., Averin, I. A., Moshnikov, V. A., Nalimova, S. S., Donkova, B. V., Dimitrov, D. T., Georgieva, A. T. A percolation model of semiconductor gas sensors with a hierarchical pore structures. EAI Endorsed Transactions on Energy Web 2019, 6(22), e10.].

The manifestation of certain polarization phenomena is strongly dependent on the technological conditions for the obtaining of porous silicon. However, according to the authors opinion, in order to better understand the physics of processes, it is necessary to highlight the dominant, most clearly manifested mechanisms of polarization.

We have corrected the text of the manuscript, considering the remarks: we changed text (lines 177-191 and  208-221), excluded 202-204.

The authors would like to thank Reviewer 3 for a detailed discussion of the results. In the process of preparing the answers for Reviewer’s comments, new ideas appeared on the continuation of research on this topic, taking into account the influence of the microstructure of porous silicon and mechanical stresses on heterointerfaces upon its electrophysical properties.

Yulia Spivak
